# Use of Daptomycin to Manage Severe MRSA Infections in Humans

**DOI:** 10.3390/antibiotics14060617

**Published:** 2025-06-18

**Authors:** Marco Fiore, Aniello Alfieri, Daniela Fiore, Pasquale Iuliano, Francesco Giuseppe Spatola, Andrea Limone, Ilaria Pezone, Sebastiano Leone

**Affiliations:** 1Department of Women, Child and General and Specialized Surgery, University of Campania “Luigi Vanvitelli”, 80138 Naples, Italy; anielloalfieri@gmail.com; 2Department of Elective Surgery, Postoperative Intensive Care Unit and Hyperbaric Oxygen Therapy, A.O.R.N. Antonio Cardarelli, V.le Antonio Cardarelli 9, 80131 Naples, Italy; 3Anesthesia and Intensive Care Unit, ASUR Marche, Area Vasta n. 4, Via Murri, 63023 Fermo, Italy; daniela.fiore@sanita.marche.it; 4Department of Internal Medicine, Infectious Diseases Section, San Giuseppe Moscati Hospital, Contrada Amoretta, 83100 Avellino, Italy; pasquale.iuliano@aornmoscati.it (P.I.); francescogiuseppe.spatola@studenti.unicampania.it (F.G.S.); andrea.limone@unicampania.it (A.L.); sebastianoleone@yahoo.it (S.L.); 5Department of Mental Health and Public Medicine, Infectious Diseases Section, University of Campania “L. Vanvitelli”, 80138 Naples, Italy; 6Department of Pediatrics, “San Giuseppe Moscati” Hospital, 81031 Aversa, Italy; ipezone@yahoo.it

**Keywords:** methicillin-resistant *Staphylococcus aureus*, daptomycin, antibiotic resistance, combination therapy, bacteremia

## Abstract

Methicillin-resistant *Staphylococcus aureus* (MRSA) represents a major therapeutic challenge due to its multidrug-resistance and the associated clinical burden. Daptomycin (DAP), a cyclic lipopeptide antibiotic, has become a key agent for the treatment of severe MRSA infections owing to its rapid bactericidal activity and favourable safety profile. In this narrative review, we examine studies published between 2010 and April 2025. The data suggest that treatment with high-dose (8–10 mg kg⁻^1^) DAP shortened the time to blood-culture sterilisation by a median of 2 days compared with standard-dose vancomycin without increasing toxicity when model-informed area-under-the-curve monitoring was employed. Particular attention is given to the synergistic effects of DAP combined with fosfomycin or β-lactams, especially ceftaroline and ceftobiprole, in overcoming persistent and refractory MRSA infections; this approach results in a reduction in microbiological failure relative to monotherapy. Resistance remains uncommon (<2% of isolates), but recurrent mutations in *mprF*, *liaFSR*, and *walK* underscore the need for proactive genomic surveillance. Despite promising preclinical and clinical evidence supporting combination strategies, further randomized controlled trials are necessary to establish their definitive role in clinical practice, as are head-to-head cost-effectiveness evaluations. DAP remains a critical option in the evolving landscape of MRSA management, provided its use is integrated with precision dosing, resistance surveillance, and antimicrobial-stewardship frameworks.

## 1. Introduction

Methicillin-resistant *Staphylococcus aureus* (MRSA) remains one of the most significant threats in clinical medicine, particularly within hospital and healthcare-associated environments [1]. Through the acquisition of the *mecA* gene and additional resistance determinants, MRSA is now effectively impervious to the entire β-lactam class—from the prototypical penicillinase-stable agent methicillin to the most recent anti-staphylococcal cephalosporins—thereby obliging clinicians to rely on non-β-lactam alternatives for definitive therapy [2]. Historically, glycopeptides such as vancomycin (VAN) and teicoplanin have been considered the cornerstone therapies for MRSA infections [3]. Nevertheless, significant pharmacokinetic and pharmacodynamic limitations associated with these agents—including poor tissue penetration, slow bactericidal activity, and high protein binding—have progressively undermined their clinical efficacy [4].

Even in the context of adequate serum concentrations, VAN has suboptimal penetration into deep-seated infections such as endocarditis, osteomyelitis, and prosthetic device-related infections, and this often results in treatment failure [5]. Furthermore, the emergence of strains with intermediate susceptibility to glycopeptides, including VAN-intermediate *S. aureus* (VISA) and heterogeneous VISA (hVISA), has further complicated the choice of therapeutic strategies [6]. Notably, even in cases in which MRSA strains exhibit in vitro susceptibility to VAN, elevated minimum inhibitory concentrations (MICs) within the susceptible range (i.e., MICs ≥ 1.5 mg/L) have been associated with poorer clinical outcomes, including treatment failure and higher mortality rates [7].

These limitations have accelerated the search for alternative therapeutic agents with enhanced bactericidal properties, improved pharmacokinetics, and better tissue penetration. Daptomycin (DAP), a cyclic lipopeptide antibiotic first approved by the U.S. Food and Drug Administration (FDA) in 2003, has emerged as a promising solution for the management of complicated skin and skin-structure infections (cSSSIs), right-sided infective endocarditis, and bloodstream infections caused by MRSA and other Gram-positive pathogens [8].

DAP exerts its rapid bactericidal activity through a unique mechanism of action: it binds calcium ions, inserts into the bacterial cell membrane, and causes rapid depolarization, leading to inhibition of protein, DNA, and RNA synthesis and subsequently to cell death [9]. Importantly, DAP does not cause lysis of bacterial cells, potentially minimizing the release of inflammatory mediators and contributing to its favourable safety profile [10].

This narrative review aims to comprehensively evaluate the role of DAP in the management of serious MRSA infections. We explore the microbiological and pharmacological characteristics of DAP, examine the clinical efficacy of DAP in monotherapy and combination regimens, discuss emerging resistance mechanisms, and assess the therapeutic strategies used to overcome resistance, particularly those involving combination with agents such as fosfomycin (FOS) and β-lactams.

For this updated narrative review, we conducted a structured search of PubMed, EMBASE, Cochrane CENTRAL, Web of Science, and Scopus for studies published between 1 January 2010 and 15 April 2025. The search combined the terms “daptomycin”, “MRSA”, “methicillin-resistant Staphylococcus aureus”, “combination therapy”, “pediatric”, and “therapeutic drug monitoring”. Only full-text articles in English were considered.

## 2. MRSA—Clinical Impact

MRSA infections are associated with substantial clinical and economic burdens, including increased mortality, longer hospital stays, and elevated healthcare costs [11]. In the case–control studies cited below, the odds ratio (OR) expresses how strongly the exposure (MRSA infection) is associated with an outcome relative to the comparator group; an OR > 1 denotes higher odds of the outcome, whereas an OR < 1 denotes lower odds. The accompanying 95% confidence interval (CI) provides the range within which the true OR is likely to lie; if the CI excludes 1, the association is considered statistically significant. Each study’s primary objective was to quantify the excess risk or resource use attributable to MRSA versus methicillin-susceptible *S. aureus* (MSSA). Engemann et al. demonstrated that patients with surgical-site infections due to MRSA had a higher 90-day mortality rate compared to those with methicillin-susceptible *S. aureus* (MSSA) infections (odds ratio [OR] 3.4, 95% confidence interval [CI] 1.5–7.2) [12]. Additionally, MRSA infections were associated with a significant increase in hospital charges and length of stay compared to MSSA infections [12]. Hirabayashi et al. reported that MRSA infections resulted in a 21% longer hospital stay (OR 1.21, 95% CI 1.03–1.42) and a 70% higher cost of hospitalization (OR 1.70, 95% CI 1.39–2.07) compared to MSSA infections [13]. 

Age-stratified analyses reveal important differences: in a multicentre US cohort of 232 children hospitalised with MRSA bacteraemia, treatment failure occurred in 31% of episodes and 30-day mortality was 2% [14], while a meta-analysis of seven paediatric and neonatal studies demonstrated that MRSA doubled the odds of death compared with MSSA (pooled OR 2.33, 95% CI 1.42–3.82) [15]. Furthermore, in the neonatal ICU cohort described by Song et al. (2010), MRSA infection—compared with mere MRSA colonisation—independently prolonged hospital stay by 40 days (95% CI 34.2–45.6) and added USD 164 301 to hospital charges (95% CI USD 158 712–USD 169 889) [16]. Conversely, adults aged ≥ 65 years experience the highest absolute mortality—reported to be 17% and 30% in contemporary series—with MRSA bacteraemia conferring an almost two-fold increased risk of death versus MSSA (OR 1.93, 95% CI 1.54–2.42) [17,18,19]. The mortality associated with MRSA infections is notably higher compared to that associated with MSSA infections. A meta-analysis conducted by Whitby et al. demonstrated that MRSA bacteraemia was associated with approximately a twofold increased risk of death compared with MSSA bacteraemia (relative risk [RR] 2.12, 95% CI 1.76–2.57 using a fixed-effects model; RR 2.03, 95% CI 1.55–2.65 using a random-effects model) [17]. Cosgrove et al. also confirmed these findings, reporting an OR of 1.93 (95% CI 1.54–2.42) for mortality in MRSA bacteraemia compared with MSSA bacteraemia [18]. More recently, a large systematic review and meta-analysis by Bai et al. involving 341 studies found that the overall mortality rates for *S. aureus* bacteraemia were 10.4% at 7 days, 13.3% at 2 weeks, 18.1% at 1 month, 27.0% at 3 months, and 30.2% at 1 year [19]. The meta-regression analysis indicated that higher proportions of MRSA among isolates were associated with increased 1-month mortality (adjusted OR 1.04, 95% CI 1.02–1.06 per 10% increase in MRSA proportion) [19]. Taken together, these data underscore the disproportionate clinical and economic impact of MRSA across age groups—particularly in neonates and children—and reinforce the need for optimised, age-specific therapeutic strategies.

## 3. Daptomycin—Microbiology and Resistance

DAP is a cyclic lipopeptide antibiotic with potent bactericidal activity against Gram-positive pathogens, including MRSA, VISA, and vancomycin-resistant enterococci (VRE) [20]. As detailed in the Introduction, its calcium-mediated insertion into the bacterial membrane triggers rapid depolarisation; here, we focus instead on the pharmacodynamic consequences—markedly concentration-dependent killing, a prolonged post-antibiotic effect, and superior penetration of staphylococcal biofilms—that collectively underpin its clinical utility against high-burden or device-related MRSA infections [21,22]. In contrast to glycopeptides, DAP penetrates biofilms more effectively, making it an attractive agent for the treatment of biofilm-associated infections, such as prosthetic-device-related infections and endocarditis [23]. Despite its excellent in vitro and in vivo efficacy, clinical reports of emerging DAP resistance have surfaced, particularly in the context of prolonged therapy for bacteraemia and endocarditis [24]. Resistance to DAP typically occurs via mutations in genes associated with cell-membrane charge and fluidity, including *mprF*, *liaFSR*, and *yycG* [25,26]. Mutations in *mprF* result in increased lysyl-phosphatidylglycerol production and incorporation into the bacterial membrane, producing a more positively charged surface that repels the positively charged DAP–calcium complex [27]. Mutations in *liaFSR* (a three-component regulatory system) and *yycG* (part of the WalKR two-component system) are associated with cell-envelope stress responses and alterations in membrane fluidity and thickness, further contributing to reduced DAP susceptibility [28]. In Figure 1, there is a schematic illustration of mutations that reduce susceptibility or cause resistance to DAP. Cross-resistance phenomena between DAP and glycopeptides have also been observed, particularly in strains with thickened cell walls and altered membrane charges [29]. The development of DAP resistance often correlates with an increase in VAN MIC, even in strains still categorized as VAN-susceptible, emphasizing the need for vigilant therapeutic monitoring [30]. Strategies to overcome or prevent DAP resistance include optimizing the DAP dose (often ≥8–10 mg/kg/day) and employing combination therapies with agents that synergistically enhance DAP activity, such as FOS or β-lactams [31]. Such combination approaches aim not only to increase bactericidal activity but also to prevent the emergence of DAP-resistant subpopulations during therapy [32].

## 4. Clinical Use—Daptomycin Monotherapy

### 4.1. Critical Appraisal of Recent Evidence and Guideline Updates

The FDA approved the use of DAP for the treatment of cSSSIs and *S. aureus* bloodstream infections, including right-sided infective endocarditis [33]. Its potent, rapid, and concentration-dependent bactericidal activity makes it an attractive therapeutic option for MRSA infections, particularly when VAN MIC is elevated within the susceptible range [34].

In adult cohorts, high-dose daptomycin (median 9.8 mg kg⁻^1^ day⁻^1^, range 8.2–10.0 mg kg⁻^1^) achieved microbiological eradication in 89% and overall clinical success in 86% of cases, with *no* treatment discontinuations due to elevated creatine phosphokinase levels; these results thereby provide evidence for the safety and effectiveness of regimens using ≥ 8 mg kg⁻^1^ for the treatment of adults [35].

Several clinical trials and observational studies have confirmed the efficacy of DAP monotherapy for serious MRSA infections; see Table 1 below. Fowler et al. demonstrated in a pivotal randomized clinical trial that DAP was non-inferior to standard therapy (VAN or anti-staphylococcal penicillins plus gentamicin) for the treatment of *S. aureus* bacteraemia and right-sided endocarditis [36]. DAP-treated patients exhibited comparable clinical success rates with fewer nephrotoxic adverse events compared to those receiving standard therapy.

Clinical data show that high-dose daptomycin retains both efficacy and safety in patient groups traditionally considered ‘high-risk’. In adults ≥ 65 years enrolled in an open-label, multicentre, phase IIIb trial of first-line therapy for complicated skin-and-soft-tissue infection, daptomycin produced high clinical success with no excess treatment discontinuations or drug-related myopathies, confirming that advanced age does not mandate dose attenuation [37]. Similarly, a 13-centre cohort of hospitalised obese adults (median BMI ≈ 34 kg m⁻^2^) who received predominantly ≥ 8 mg kg⁻^1^ day⁻^1^ demonstrated cure rates comparable to those seen in non-obese controls and a very low incidence (~2%) of creatine-phosphokinase elevations requiring discontinuation, supporting weight-based high-dose regimens without arbitrary capping in obesity [38].

**Table 1 antibiotics-14-00617-t001:** Major studies on daptomycin monotherapy for MRSA infections.

Study	Year	Study Design	Population	Main Findings
Fowler et al. [36]	2006	Randomized controlled trial	Adults with MRSA bacteraemia and right-sided endocarditis	DAP non-inferior to standard therapy, fewer nephrotoxic events
Murray et al. [39]	2013	Observational study	Adults with MRSA bacteraemia	Faster clearance of bacteraemia with DAP compared to VAN
Kullar et al. [40]	2013	Observational cohort	Critically ill patients with MRSA bacteraemia	DAP associated with higher rates of microbiologic success

Moreover, DAP-treated patients exhibit faster clearance of bacteraemia compared to VAN-treated patients [39], with higher rates of microbiologic success [40]; real-world observational studies have shown that DAP monotherapy leads to faster clearance of MRSA bacteraemia compared to VAN monotherapy, especially in cases in which the VAN MIC is ≥1.5 mg/L [41]. Additionally, DAP therapy is associated with a lower risk of nephrotoxicity, an important consideration for critically ill patients who are often receiving multiple nephrotoxic agents [18]. Despite these advantages, treatment failures with DAP monotherapy have been reported, particularly in patients with high-bacterial-burden infections such as left-sided endocarditis or prosthetic-device-associated infections [32]. Suboptimal dosing (≤6 mg/kg/day) and delayed initiation of DAP have been identified as risk factors for treatment failure and the emergence of DAP resistance [42]. Accordingly, experts recommend higher doses of DAP (8–10 mg/kg/day) for serious MRSA infections to maximize bactericidal activity and prevent resistance [43]. The guidelines of the Infectious Diseases Society of America (IDSA) on MRSA bacteraemia, as last updated in 2018, recommend DAP 6 mg/kg/day (A-I) and doses 8–10 mg/kg as expert advice for complicated infections (B-III), but do not make this a standard recommendation [44]. Although observational data consistently show more rapid blood-culture sterilization with high-dose daptomycin, two recent propensity-matched studies found no mortality difference versus standard-dose vancomycin once adequate source control was obtained [41,45]. In a recent meta-analysis, it was found that switching early from vancomycin to DAP was significantly associated with lower mortality odds for patients with MRSA bloodstream infections [46].

These conflicting findings underscore the importance of individual PK/PD optimization rather than blanket escalation.

### 4.2. Pharmacokinetic Advances and TDM

Population PK studies in critically ill adults show a 40–60% rise in daptomycin clearance during the first 72 h of septic shock [47,48,49].

Real-time Bayesian therapeutic drug monitoring (TDM) platforms that couple sparse sampling with Monte Carlo simulations have lowered the rates of nephrotoxicity and CPK elevation in small single-centre cohorts [50,51,52]. It is recommended to aim for an AUC_0–24_ of 550–900 mg·h/L and a trough ≤24 mg/L, as values above this threshold are associated with a ≥50% risk of CPK elevation [53].

Therefore, TDM of DAP has been proposed to optimize outcomes, particularly in patients with altered pharmacokinetics, such as those with obesity, critical illness, or renal dysfunction [54]. Furthermore, when high-bacterial-density infections or DAP non-susceptibility are suspected, combination-therapy strategies should be considered to enhance bactericidal activity and prevent resistance development [55].

## 5. Clinical Use—Daptomycin Plus Fosfomycin

The combination of DAP and FOS has gained increasing attention as a therapeutic strategy for MRSA infections, particularly in cases of persistent bacteraemia or emerging DAP resistance [56]. Adult pharmacokinetic data show that a FOS regimen of 4 g every 6 h (renally adjusted) maintains unbound plasma concentrations above a MRSA MIC of 32 mg L⁻^1^ for ≥ 70% of the dosing interval, thereby providing sustained peptidoglycan-pathway inhibition that complements the rapid, concentration-dependent membrane depolarisation produced by DAP at doses of 8–10 mg kg⁻^1^ day⁻^1^ [57].

The rationale behind this combination lies in their complementary mechanisms of action: while DAP disrupts the bacterial membrane, FOS inhibits the early stages of peptidoglycan biosynthesis, potentially increasing membrane permeability and facilitating DAP’s bactericidal action [58].

In vitro studies have demonstrated significant synergistic activity between DAP and FOS against MRSA strains, including those with reduced susceptibility to glycopeptides and lipopeptides [59]. Time–kill assays revealed that the combination leads to faster and more profound bacterial killing compared to monotherapy with either agent alone [60]. Aktas and Derbentli evaluated the in vitro efficacy of DAP in combination with various agents and found that synergy with FOS occurred in 100% of MRSA isolates tested [58].

Animal models have provided further support for the efficacy of the DAP–FOS combination. In a rabbit model of experimental endocarditis caused by MRSA, García-de-la-Mària et al. demonstrated that DAP–FOS combination therapy significantly increased the proportion of sterile vegetations and reduced bacterial densities compared to DAP monotherapy [59]. Moreover, DAP–FOS combination therapy had efficacy to those of regimens with high-dose DAP (10 mg/kg/day) and DAP plus cloxacillin.

Mishra et al. investigated the impact of DAP–FOS combinations against DAP-susceptible and DAP-resistant MRSA strains using in vitro, ex vivo, and in vivo models. The study showed that the combination enhanced bacterial clearance, prevented the emergence of DAP resistance, and resensitised DAP-resistant strains to a DAP-susceptible phenotype, emphasizing the potential of DAP–FOS in combating DAP resistance [60].

Clinically, observational studies and case series have reported favourable outcomes with DAP–FOS combination therapy in persistent MRSA bacteraemia [61]. Miró et al. found that DAP–FOS combination therapy had synergistic and bactericidal activity against most MRSA isolates in vitro, supporting its clinical use [56]. In a retrospective cohort study, Coronado-Álvarez et al. documented the use of DAP–FOS combination therapy in patients with persistent staphylococcal bacteraemia, demonstrating improved microbiological outcomes compared to standard therapies [62].

Furthermore, the randomized, open-label BACSARM trial compared high-dose DAP monotherapy to DAP–FOS combination therapy in patients with MRSA bacteraemia [31]. Although differences in the primary endpoint (treatment success at 6 weeks) did not reach statistical significance, combination therapy was associated with significantly lower rates of microbiologic failure and reduced rates of complicated bacteraemia.

Overall, the accumulating preclinical and clinical evidence suggests that DAP–FOS combination therapy is a promising option for the management of serious MRSA infections, particularly in cases complicated by high bacterial burdens, persistent bacteraemia, or DAP non-susceptibility; see Table 2 below.

## 6. Clinical Use—Daptomycin Plus Β-Lactams

The combination of DAP with β-lactam antibiotics has emerged as another promising therapeutic strategy to enhance bacterial killing and prevent resistance in MRSA infections [63]. This synergistic interaction is largely explained by the so-called “seesaw effect,” where MRSA strains with reduced susceptibility to glycopeptides or lipopeptides exhibit increased susceptibility to β-lactams [64].

Several β-lactams, including oxacillin, ceftaroline (CPT), ceftobiprole (BPR), and even traditional anti-staphylococcal penicillins, have demonstrated synergy with DAP against MRSA in vitro and in animal models [32,63,65,66].

DAP–CPT combination therapy restores DAP membrane binding and bactericidal activity, even against DAP-non-susceptible strains [67].

Werth et al. evaluated the DAP–CPT combination in a pharmacokinetic/pharmacodynamic model simulating endocardial vegetations and observed significantly enhanced bacterial killing compared to that observed with either agent alone [67]. Moreover, exposure to CPT led to enhanced DAP-induced membrane depolarization and increased susceptibility to host immune defences [67].

In a hollow-fibre infection model, Barber et al. demonstrated that the DAP–CPT combination resulted in robust bactericidal activity, allowing for the potential use of de-escalation strategies once microbiologic clearance was achieved [43]. Importantly, this combination was also shown to be effective against biofilm-producing, DAP-non-susceptible MRSA strains [68].

Clinical evidence supporting the use of DAP–β-lactam combination therapy is accumulating. Sakoulas et al. reported successful salvage therapy using DAP–CPT combination therapy in patients with persistent MRSA bacteraemia refractory to standard treatments [69]. A pilot randomized clinical trial by Geriak et al. suggested that early initiation of DAP–CPT combination therapy may reduce mortality compared to standard therapy, although limitations in study design precluded definitive conclusions [70].

Subsequent observational studies and retrospective cohorts have reinforced the potential utility of DAP–CPT combination therapy, particularly when it is initiated early in the course of infection [71,72,73]. Notably, combination therapy has been associated with a shorter duration of bacteraemia, reduced risk of persistent infection, and lower recurrence rates compared to monotherapy [74].

Meta-analyses by Yi et al. and Huang et al. indicated that while DAP–β-lactam combination therapies may not significantly reduce mortality compared to standard therapies, they were associated with microbiological advantages, including faster clearance of bacteraemia and lower recurrence rates [74,75].

The combination of DAP with BPR has also shown promise. Barber et al. demonstrated in vitro that the DAP-BPR combination resulted in a four-fold reduction in DAP MICs and synergistic killing of MRSA strains [43]. More recently, Jorgensen et al. observed that the addition of beta-lactams to DAP was associated with improved clinical outcomes in patients with MRSA BSI [76].

Clinical reports on DAP–BPR combination therapy remain limited to case series and observational experiences but suggest potential efficacy in difficult-to-treat infections, including prosthetic valve endocarditis [77,78].

Overall, the DAP–β-lactam-combination strategy represents a valuable option for managing refractory MRSA infections, particularly in cases involving persistent bacteraemia, high bacterial loads, or reduced DAP susceptibility (see Table 3 below). However, larger, well-designed, randomized controlled trials are still needed to establish the superiority of combination therapy over monotherapy definitively.

## 7. Clinical Use in Paediatric Patients

While DAP has been widely adopted for use in adult populations since its approval, its application in paediatric patients has garnered increasing attention in recent years [71]. Initial clinical trials evaluated the safety and efficacy of DAP in children aged 1 to 17 years with CSIS [80]. Bradley et al. demonstrated that DAP was non-inferior to standard-of-care antibiotics for CSIS, with a favourable safety profile and similar rates of clinical success [80]. Further studies extended the use of DAP to paediatric patients with *S. aureus* bacteraemia and osteomyelitis. Arrieta et al. reported that DAP was effective in treating staphylococcal bloodstream infections in children, with outcomes comparable to those achieved with VAN and a low incidence of adverse events [81]. Subsequently, DAP was shown to be effective in the treatment of acute hematogenous osteomyelitis in paediatric populations, achieving high cure rates and minimal toxicity [82]. In real-world clinical settings, DAP has been primarily utilized for the management of serious Gram-positive infections in children, including central-line-associated bloodstream infections, endocarditis, surgical-site infections, and osteomyelitis [83]. Retrospective studies have confirmed its efficacy and tolerability, supporting its role as an alternative when standard therapies are contraindicated or ineffective. Pharmacokinetic (PK) studies have highlighted important differences in DAP metabolism between paediatric patients and adults. Children demonstrate increased DAP clearance rates, necessitating age-adjusted dosing strategies to achieve therapeutic exposures like those observed in adults [84]. Using pharmacokinetic modelling and Monte Carlo simulations, Olney et al. recommended daily DAP doses of at least 20 mg/kg for children aged ≤6 years, 17 mg/kg for children 7–11 years, and 10 mg/kg for adolescents aged ≥12 years to optimize pharmacodynamic targets [84]. Although paediatric experience with DAP is expanding, careful attention to dosing regimens and monitoring for potential musculoskeletal adverse events, such as myopathy, remain essential. Nonetheless, DAP offers a valuable therapeutic option for serious MRSA infections in paediatric patients, especially when standard treatments are limited by toxicity, resistance, or intolerance.

## 8. Discussion

DAP has become indispensable in the management of severe MRSA infections, but the evidence base is heterogeneous and often conflicting. The present review therefore shifts from a descriptive to a critical stance, analysing how study design, dosing, timing of therapy, and microbiological variables shape clinical outcomes.

### 8.1. Monotherapy

The pivotal randomised trial by Fowler et al. established the non-inferiority of DAP 6 mg kg⁻^1^ day⁻^1^ versus standard therapy but enrolled mostly patients with uncomplicated bacteraemia or right-sided endocarditis and still recorded appreciable microbiological failure and emergent resistance [36]. Contemporary comparative-effectiveness analyses reveal that an early (≤72 h) switch from vancomycin to DAP is associated with lower 30-day mortality, whereas a delayed switch confers no benefit [45]. A 41-study meta-analysis published in 2024 reports a non-significant mortality odds ratio of 0.81 in favour of DAP, with substantial heterogeneity (I^2^ = 63%) that is largely explained by dose, switch timing, and vancomycin MIC [46]. These data suggest that DAP efficacy is modulated by pharmacodynamic and clinical context; routine escalation to a high dose should therefore be guided by individual PK/PD considerations rather than blanket rules.

### 8.2. Combination Regimens

Synergy studies suggest that adding fosfomycin (FOS) or a β-lactam can accelerate bacterial killing and suppress resistance, yet robust clinical evidence for this step remains scarce. The BACSARM trial of DAP + FOS reduced microbiological failure but failed to meet its primary composite endpoint and was limited by its open-label design and fixed 6 mg kg⁻^1^ DAP dose [31]. Evidence for DAP paired with advanced cephalosporins is dominated by a small pilot RCT [70] and an IPTW-adjusted cohort by Jorgensen et al. [76]. Both hint at benefit but were weakened by early termination or residual confounding. Two meta-analyses confirm faster clearance of bacteraemia without a consistent mortality benefit, and the overall GRADE certainty of evidence remains low. Clinicians should balance these potential advantages against added toxicity and cost, reserving combination therapy for persistent bacteraemia, high-inoculum infections, or suspected DAP non-susceptibility.

### 8.3. Special Settings and Pharmacokinetics

Population pharmacokinetic analyses show a 40–60% increase in DAP clearance during the first 72 h of septic shock, raising the risk of under-exposure in critically ill patients [47,48,49]. Real-time Bayesian AUC-guided dosing platforms have lowered both nephrotoxicity and CPK elevation in small cohorts, supporting the adoption of therapeutic-drug monitoring (TDM) where available [50,51,52]. When vancomycin MICs exceed 1 mg L⁻^1^ or high bacterial burdens are present, early initiation of treatment with high-dose DAP (8–10 mg kg⁻^1^) with TDM appears prudent [42].

### 8.4. Limitations of Current Evidence and Future Directions

Much of the literature comprises retrospective cohorts prone to immortal-time bias and exposure misclassification; endpoints vary from 14-day mortality to “microbiological success,” hampering synthesis. Registry datasets such as EU-CORE lack internal controls and should inform hypotheses rather than practice [34]. Future directions for research are as follows. Two phase-III trials are evaluating DAP + ceftaroline for complicated left-sided endocarditis, and another is testing DAP 10–12 mg kg⁻^1^ versus 6 mg kg⁻^1^ in high-inoculum infections. A harmonised core outcome set for MRSA bacteraemia and wider access to AUC-guided TDM are essential to resolve the remaining uncertainties.

## 9. Conclusions

DAP remains a cornerstone in the management of severe MRSA infections due to its potent, rapid bactericidal activity, favourable safety profile, and efficacy against biofilm-associated infections. Its application is particularly valuable in cases in which the VAN MIC is elevated or where treatment failure with standard therapies has occurred. Translating this microbiological potency into consistent clinical benefit now depends on overcoming three pragmatic hurdles: (i) timely attainment of target exposures through model-informed precision dosing, (ii) systematic surveillance for emergent daptomycin non-susceptibility, and (iii) demonstration of economic value versus newer lipoglycopeptides or combination regimens. Emerging resistance mechanisms, although still relatively rare, highlight the need for optimized dosing strategies and careful therapeutic monitoring. Combination regimens, particularly those incorporating FOS or β-lactams such as CPT or BPR, have shown promising synergistic effects both in vitro and in clinical settings. These combinations not only enhance bactericidal activity but also appear to prevent or reverse DAP resistance, thereby diversifying the therapeutic arsenal against refractory MRSA infections. The use of DAP in paediatric populations has also expanded, with this drug demonstrating favourable efficacy and safety profiles when appropriately dosed. Cost–utility analyses indicate that early, high-dose use of daptomycin—alone or in β-lactam combinations—is cost-effective when the vancomycin MIC is ≥ 1 mg L⁻^1^ or when the risk of acute kidney injury is high.

Despite these advances, larger randomized controlled trials are necessary to definitively establish the role of DAP-based combination therapies and to optimize treatment protocols across diverse patient populations. Future investigations should adopt adaptive platform designs comparing model-informed versus fixed dosing, embed pharmacoeconomic endpoints, and incorporate province-wide genomic resistance surveillance to capture emergent *mprF*, *liaFSR*, and *walK* polymorphisms.

In conclusion, the optimal use of DAP demands attention to dose, timing, combination partners, and patient-specific PK/PD. By explicitly acknowledging the heterogeneity and limitations of existing studies, this review provides a more reliable framework for clinical decision-making and, by integrating PK/PD, resistance biology and health-economic considerations and also offers a sustainable blueprint for daptomycin stewardship in the era of escalating antimicrobial resistance.

## Figures and Tables

**Figure 1 antibiotics-14-00617-f001:**
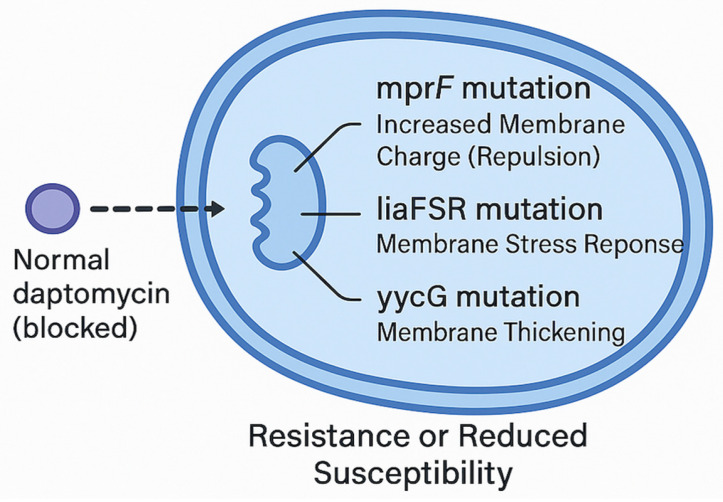
Mutations that reduce susceptibility or cause resistance to daptomycin.

**Table 2 antibiotics-14-00617-t002:** Studies evaluating daptomycin plus fosfomycin.

Study	Year	Study Design	Population	Main Findings
García-de-la-Mària et al. [59]	2018	Rabbit endocarditis model	Experimental MRSA infection	DAP+FOS combination improved bacterial clearance
Mishra et al. [60]	2022	In vitro, ex vivo, in vivo models	DAP-susceptible and DAP-resistant MRSA	Synergy, resensitisation of resistant strains
Coronado-Álvarez et al. [62]	2019	Retrospective cohort	Patients with persistent bacteraemia	DAP+FOS associated with improved microbiologic outcomes
BACSARM trial [31]	2021	Randomized controlled trial	MRSA bacteraemia patients	DAP+FOS lowered microbiological failure rates

**Table 3 antibiotics-14-00617-t003:** Studies evaluating daptomycin plus β-lactams.

Study	Year	Study Design	Population	Main Findings
Werth et al. [79]	2013	Pharmacokinetic/pharmacodynamic model	Simulated endocardial vegetations	DAP+CPT enhanced bactericidal activity
Sakoulas et al. [69]	2014	Case series	Persistent MRSA bacteraemia	Successful salvage therapy with DAP+CPT
Geriak et al. [70]	2019	Pilot randomized trial	MRSA bacteraemia	Trend towards lower mortality with early use of DAP+CPT
Barber et al. [79]	2014	In vitro study	MRSA strains	DAP+BPR combination reduced MIC and enhanced killing

## Data Availability

Data are available on reasonable request to the corresponding author.

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
