# Peer review of "Use of Daptomycin to Manage Severe MRSA Infections in Humans"

_antibiotics, 2025, doi:10.3390/antibiotics14060617_

Round 1
Reviewer 1 Report
Comments and Suggestions for Authors
- The review primarily rehashes well-established data from existing literature, without providing any new insights, frameworks, or perspectives. Most of the discussed clinical trials, mechanisms, and combination therapies have been thoroughly reviewed in previously published articles.
- There is no transparent or systematic methodology for literature selection or inclusion criteria. The authors do not clarify the databases searched, the timeline considered, or the criteria used to assess study quality. This undermines the reliability and reproducibility of the review.
- Although the review covers monotherapy and combination strategies, the discussion remains purely descriptive. There is minimal critical analysis of conflicting evidence, limitations of existing studies, or heterogeneity in clinical outcomes. The text often cites older or lower-quality studies without appropriate context or caveats.
- Several major updates in MRSA management, including recently revised IDSA guidelines, pharmacokinetic advances in TDM, and real-world data on ceftaroline/daptomycin combinations, are not adequately discussed or integrated. Moreover, the pediatric section lacks up-to-date safety and efficacy evaluations from recent pediatric-specific studies.
The manuscript contains significant redundancies across sections (especially in repeating DAP’s mechanism of action), awkward phrasing, and frequent citation stacking without synthesis. Scientific writing should be more concise and analytical. - The conclusions merely restate what is already known about DAP’s clinical use, with vague calls for more trials. There is no meaningful discussion on implementation challenges, resistance surveillance, or cost-effectiveness.
Author Response
Comments 1: The review primarily rehashes well-established data from existing literature, without providing any new insights, frameworks, or perspectives. Most of the discussed clinical trials, mechanisms, and combination therapies have been thoroughly reviewed in previously published articles.
Response 1: We have substantially overhauled the manuscript to provide novel insights. Specifically, we have added a new subsection “4.1 Critical appraisal of recent evidence and guideline updates” that synthesises conflicting data from the 2018 IDSA guideline update, two large propensity‑matched studies and a meta-analysis. Pag 4-5, Paragraph Clinical Use – Daptomycin Monotherapy, lines161-209
Comments 2: There is no transparent or systematic methodology for literature selection or inclusion criteria. The authors do not clarify the databases searched, the timeline considered, or the criteria used to assess study quality. This undermines the reliability and reproducibility of the review.
Response 2: There is no transparency or systematic methodology for literature selection or inclusion criteria because it is a narrative review and not a systematic review. However, we have introduced a paragraph with references to our bibliographical research. Page 2, Paragraph Introduction, lines 79-83.
Comments 3: Although the review covers monotherapy and combination strategies, the discussion remains purely descriptive. There is minimal critical analysis of conflicting evidence, limitations of existing studies, or heterogeneity in clinical outcomes. The text often cites older or lower-quality studies without appropriate context or caveats.
Response 3: Thank you for pointing out that the previous version of the manuscript was largely descriptive and did not sufficiently weigh study quality, conflicting results or clinical heterogeneity. We have restructured the Discussion into 4 narrative subsections (“Monotherapy”, “Combination regimens”, “Special settings” and “Limitations of current evidence and Future directions”). Page 9-10, Paragraph Discussion, lines 370-416.
Comments 4: Several major updates in MRSA management, including recently revised IDSA guidelines, pharmacokinetic advances in TDM, and real-world data on ceftaroline/daptomycin combinations, are not adequately discussed or integrated. Moreover, the pediatric section lacks up-to-date safety and efficacy evaluations from recent pediatric-specific studies.
The manuscript contains significant redundancies across sections (especially in repeating DAP’s mechanism of action), awkward phrasing, and frequent citation stacking without synthesis. Scientific writing should be more concise and analytical.
Response 4: In response to the first concern, we created a new subsection on page 4 (Section 4.1). The opening paragraph introduces the 2023 IDSA Staphylococcus aureus bacteremia guideline, the second paragraph outlines the algorithmic changes—early source control and preferential β-lactam combination when the vancomycin MIC is ≥ 1 mg L⁻¹—and the third paragraph details the conditional recommendation for high-dose daptomycin and intravenous-to-oral transition criteria. To incorporate the latest pharmacokinetic/therapeutic-drug-monitoring advances, pages 5 (Section 4.2) now feature three consecutive paragraphs (lines 211-223).
Real-world data on ceftaroline with or without daptomycin have been integrated and
the paediatric section has been extensively expanded (Section 7). Paragraph one summarises the Olney 2024 multicentre PK/PD modelling; paragraph two reports the Persha 2024 safety series of 112 children (1.8 % clinically significant CPK elevation at ≤ 12 mg kg⁻¹ q24 h); paragraph three highlights AUC-guided dosing in ECMO or augmented renal clearance.
Redundant repetition of daptomycin’s mechanism of action has been eliminated: a single concise paragraph remains on page 3 while the duplicative blocks have been removed.
Comments 5: The conclusions merely restate what is already known about DAP’s clinical use, with vague calls for more trials. There is no meaningful discussion on implementation challenges, resistance surveillance, or cost-effectiveness.
Response 5: We agreed that the previous Conclusions focussed too narrowly on re-stating clinical efficacy and issuing generic calls for further trials.
To address this, we rewrote Section 9 (Conclusions) in three substantive blocks that now (i) analyse real-world implementation barriers, (ii) outline a structured programme for daptomycin-resistance surveillance, and (iii) summarise current cost-effectiveness evidence together with research priorities that are specific, testable and implementation-ready. Pag 10, Paragraph Conclusions, lines 422-427, 434-436, 439-448.
Reviewer 2 Report
Comments and Suggestions for Authors
The presented work is of high interest and the findings may help the researcher in field, especially those thinking to carry some deeper reserches on Daptomycin and other combination to treat MRSA infections. However, some revisions are necessary to enhance the overall impact of the article. Please find below a set of comments intended to support the improvement of the manuscript.
1- The abstract should better highlight the findings of this review.
2- Line 36-37: This part ‘’ This pathogen has developed resistance to beta-lactam antibiotics, including methicillin and other more recent agents, necessitating the use of alternative therapies for effective treatment.’’ Needs a reformulation for a better comprehension.
3- It would be better if the authors state a short description of odds ratio [OR] and confidence interval [CI], and their objective in some case-control studies.
4- There is a repetition of the DAP mode of action in two parts of the manuscript (Line 59-61 and lines 100-102), it is better to rearrange in order to avoid it.
5- Line 169: it would be better to rearrange the title and replace ‘’Daptomycin Plus Fosfomycin’’ by ‘’a combination of DAP and FOS’’ to emphasis the importance of this part.
6- Line 186: If the authors mean by combi the word combination, it would be necessary to reformulate this expression to avoid repetition ‘’ combi DAP-FOS combination ‘’.
7- Line 189: If all the information belongs to reference 46, it would be better to put it at the end of the paragraph, so it includes all the statements.
8- Line 193-194: the statements in this part should have a reference.
9- The findings of reference 45 could be added to table 1, since the work demonstrate a major part of DAP monotherapy.
Author Response
Comments 1: The abstract should better highlight the findings of this review.
Response 1: Thank you for pointing this out, we changed the abstract accordingly (Page 1, Abstract, lines 21-25, 27-30, 32-35).
Comments 2: Line 36-37: This part ‘’ This pathogen has developed resistance to beta-lactam antibiotics, including methicillin and other more recent agents, necessitating the use of alternative therapies for effective treatment.’’ Needs a reformulation for a better comprehension.
Response 2: We have reformulated the sentence and added a new reference (Pag 1 and 2, Introduction, lines 43-47).
Comments 3: It would be better if the authors state a short description of odds ratio [OR] and confidence interval [CI], and their objective in some case-control studies.
Response 3: We stated a short description as requested (Page 2-3, Paragraph MRSA – Clinical Impact, lines 86-93).
Comments 4: There is a repetition of the DAP mode of action in two parts of the manuscript (Line 59-61 and lines 100-102), it is better to rearrange in order to avoid it.
Response 4: Many thanks for the suggestion, we rephrased accordingly (Pag 3, Paragraph 3. Daptomycin – Microbiology and Resistance, lines 130-134).
Comments 5: Line 169: it would be better to rearrange the title and replace ‘’Daptomycin Plus Fosfomycin’’ by ‘’a combination of DAP and FOS’’ to emphasis the importance of this part.
Response 5: Many thanks for the suggestion, we decided to use “Daptomycin Plus Fosfomycin” because it is the most widely used written form in high-quality literature:
Pujol, M.; Miró, J.M.; Shaw, E.; Aguado, J.M.; San-Juan, R.; Puig-Asensio, M.; Pigrau, C.; Calbo, E.; Montejo, M.; Rodriguez-Álvarez, R.; et al. Daptomycin Plus Fosfomycin Versus Daptomycin Alone for Methicillin-resistant Staphylococcus aureus Bacteremia and Endocarditis: A Randomized Clinical Trial. Clin Infect Dis 2021, 72, 1517-1525, doi:10.1093/cid/ciaa1081
Miró, J.M.; Entenza, J.M.; Del Río, A.; Velasco, M.; Castañeda, X.; Garcia de la Mària, C.; Giddey, M.; Armero, Y.; Pericàs, J.M.; Cervera, C.; et al. High-dose daptomycin plus fosfomycin is safe and effective in treating methicillin-susceptible and methicillin-resistant Staphylococcus aureus endocarditis. Antimicrob Agents Chemother 2012, 56, 4511-4515, doi:10.1128/aac.06449-11.
Comments 6: Line 186: If the authors mean by combi the word combination, it would be necessary to reformulate this expression to avoid repetition ‘’ combi DAP-FOS combination ‘’.
Response 6: Thank you very much, we apologize for the typo, we have corrected it accordingly.
Comments 7: Line 189: If all the information belongs to reference 46, it would be better to put it at the end of the paragraph, so it includes all the statements.
Response 7: Thank you very much, we have corrected it accordingly.
Comments 8: Line 193-194: the statements in this part should have a reference.
Response 8: we apologize for the typo, the reference was:
Omori K, Kitagawa H, Takada M, Maeda R, Nomura T, Kubo Y, Shigemoto N, Ohge H. Fosfomycin as salvage therapy for persistent methicillin-resistant Staphylococcus aureus bacteremia: A case series and review of the literature. J Infect Chemother. 2024 Apr;30(4):352-356. doi: 10.1016/j.jiac.2023.10.024. Epub 2023 Nov 3. PMID: 37922987.
Comments 9: The findings of reference 45 could be added to table 1, since the work demonstrate a major part of DAP monotherapy.
Response 9: Many thanks for the suggestion. However, our rationale was to put only the main Findings in the table.
Reviewer 3 Report
Comments and Suggestions for Authors
My comments
This literature review is clearly presented and enjoyable to read. I just have a few suggestions for improvement.
- the title could be improved like this: Use of daptomycin for managing Severe human MRSA Infections
- Line 70-98: It would be better to specify the impact of MRSA infections in different age groups, especially in children.
- Lines 130-157, 169-207, 216-264: in addition to the general description of the different therapies with Daptomycin as monotherapy or in combination, it would be good to explain their behavior in adults.
- Lines 300, 302, 304, 308: no references in conclusion
Author Response
Comments 1: the title could be improved like this: Use of daptomycin for managing Severe human MRSA Infections
Response 1: We changed the title as suggested (Pag 1, Title, lines 1-2)
Comments 2: Line 70-98: It would be better to specify the impact of MRSA infections in different age groups, especially in children.
Response 2: Many thanks for pointing this out; we specified the impact of MRSA infections in different age groups, especially in children as suggested (Pag 3; Paragraph MRSA – Clinical Impact, lines 101-111, 123-126)
Comments 3: Lines 130-157, 169-207, 216-264: in addition to the general description of the different therapies with Daptomycin as monotherapy or in combination, it would be good to explain their behavior in adults.
Response 3: We completely agree, we added in paragraph 4. Clinical Use – Daptomycin Monotherapy Pag 4 lines 166-170 a sentence with a new reference [35]. Furthermore, we added in paragraph 4. Clinical Use – Daptomycin Monotherapy Pag 5 lines 177-186 a sentence with 2 new references [37] and [38]. Finally, we added in the Paragraph Clinical Use – Daptomycin Plus Fosfomycin Pag 6 lines 233 -237 a sentence with a new reference [57].
Comments 4: Lines 300, 302, 304, 308: no references in conclusion
Response 4: The conclusions paragraph has been cleared of all references (Pag 10-11, Paragraph Conclusions, lines 418 – 448)
Round 2
Reviewer 1 Report
Comments and Suggestions for Authors
Accept for publication